# Enhancing Neural Machine Translation with Semantic Units

**Langlin Huang, Shuhao Gu[†], Zhuocheng Zhang, Yang Feng[*]**
Key Laboratory of Intelligent Information Processing,
Institute of Computing Technology, Chinese Academy of Sciences
University of Chinese Academy of Sciences
{huanglanglin21s, zhangzhuocheng20z,fengyang}@ict.ac.cn

## Abstract

Conventional neural machine translation (NMT) models typically use subwords and words as the basic units for model input and comprehension. However, complete words and phrases composed of several tokens are often the fundamental units for expressing semantics, referred to as semantic units. To address this issue, we propose a method Semantic Units for Machine Translation (SU4MT) which models the integral meanings of semantic units within a sentence, and then leverages them to provide a new perspective for understanding the sentence. Specifically, we first propose Word Pair Encoding (WPE), a phrase extraction method to help identify the boundaries of semantic units. Next, we design an Attentive Semantic Fusion (ASF) layer to integrate the semantics of multiple subwords into a single vector: the semantic unit representation. Lastly, the semantic-unit-level sentence representation is concatenated to the token-level one, and they are combined as the input of encoder. Experimental results demonstrate that our method effectively models and leverages semantic-unit-level information and outperforms the strong baselines. The code is available at https://github.com/ictnlp/SU4MT.

## 1 Introduction

Neural machine translation (NMT) (Kalchbrenner and Blunsom, 2013; Cho et al., 2014; Sutskever et al., 2014; Bahdanau et al., 2015b; Gehring et al., 2017; Vaswani et al., 2017; Zhang et al., 2023) has achieved significant success and has been continuously attracting significant attention. Currently, the mainstream language models use tokens as meaningful basic units, which are typically words or subwords, and in a few cases, characters (Xue et al., 2022). However, these tokens are not necessarily

---

[*]Corresponding author.
[†]This paper is done when Shuhao Gu studied in Institute of Computing Technology, and now he is working at Beijing Academy of Artificial Intelligence.

Figure 1: An example of how integrating semantic units helps reduce translation errors. Instead of translating each word separately, SU4MT learns a unified meaning of "picket line" to translate it as a whole.

semantic units in natural language. For words composed of multiple subwords, their complete meaning is expressed when they are combined. For some phrases, their overall meanings differ from those of any individual constituent word. Consequently, other tokens attend to each part of a semantic unit rather than its integral representation, which hinders the models from effectively understanding the whole sentence.

Recent studies have been exploring effective ways to leverage phrase information. Some works focus on finding phrase alignments between source and target sentences (Lample et al., 2018; Huang et al., 2018). Others focus on utilizing the source sentence phrase representations (Xu et al., 2020; Hao et al., 2019; Li et al., 2022a, 2023). However, these approaches often rely on time-consuming parsing tools to extract phrases. For the learning of phrase representations, averaging token representations is commonly used (Fang and Feng, 2022; Ma et al., 2022). While this method is simple, it fails to effectively capture the overall semantics of the phrases, thereby impacting the model performance. Some researchers proposed to effectively learn phrase representations (Yamada et al., 2020; Li et al., 2022b) using BERT (Devlin et al., 2019), but this introduces a large number of additional parameters.

Therefore, we propose a method for extracting semantic units, learning the representations of them, and incorporating semantic-level representations into Neural Machine Translation to enhance translation quality. In this paper, a semantic unit means a contiguous sequence of tokens that collectively form a unified meaning. In other words, a semantic unit can be a combination of subwords, words, or both of them.

For the extraction of semantic units, it is relatively easy to identify semantic units composed of subwords, which simplifies the problem to extracting phrases that represent unified meanings within sentences. Generally, words that frequently co-occurent in a corpus are more likely to be a phrase. Therefore, we propose Word Pair Encoding (WPE), a method to extract phrases based on the relative co-occurrence frequencies of words.

For learning semantic unit representations, we propose an Attentive Semantic Fusion (ASF) layer, exploiting the property of attention mechanism (Bahdanau et al., 2015b) that the length of the output vector is identical with the length of the query vector. Therefore, it is easy to obtain a fixed-length semantic unit representation by controlling the query vector. To achieve this, pooling operations are performed on the input vectors and the result is used as the query vector. Specifically, a combination of max-pooling, min-pooling, and average pooling is employed to preserve the original semantics as much as possible.

After obtaining the semantic unit representation, we propose a parameter-efficient method to utilize it. The token-level sentence representation and semantic-unit-level sentence representation are concatenated together as the input to the encoder layers, with separate positional encoding applied. This allows the model to fully exploit both levels of semantic information. Experimental results demonstrate this approach is simple but effective.

## 2 Background

### 2.1 Transformer Neural Machine Translation model

Neural machine translation task is to translate a source sentence $x$ into its corresponding target sentence $y$. Our method is based on the Transformer neural machine translation model (Vaswani et al., 2017), which is composed of an encoder and a decoder to process source and target sentences respectively. Both the encoder and decoder consist of a word embedding layer and a stack of 6 encoder/decoder layers. The word embedding layer maps source/target side natural language texts $x/y$ to their vectored representation $x_{emb}/y_{emb}$. An encoder layer consists of a self-attention module and a non-linear feed-forward module, which outputs the contextualized sentence representation of **x**. A decoder layer is similar to the encoder layer but has an additional cross-attention module between self-attention and feed-forward modules. The input of a decoder layer $y_{emb}$ attends to itself and then cross-attends to **x** to integrate source sentence information and then generate the translation $y$. The generation is usually auto-regressive, and a cross-entropy loss is applied to train a transformer model:

$$\mathcal{L}_{CE} = -\sum_t \log p(y_t|x, y_{<t}) \quad (1)$$

An attention module projects a query and a set of key-value pairs into an output, where the query, key, value, and output are vectors of the same dimension, and the query vector determines the output's length. The output is the weighted sum of values, where the weight is determined by a dot product of query and key. For training stability, a scaling factor $\frac{1}{\sqrt{d}}$ is applied to the weight. The attention operation can be concluded as equation (2)

$$Attention(Q, K, V) = softmax(\frac{QK^T}{\sqrt{d}})V \quad (2)$$

Specifically, in the transformer model, the self-attention module uses a same input vector as query and key-value pairs, but the cross-attention module in the decoder uses target side hidden states as query and encoder output **x** as key-value pairs.

### 2.2 Byte Pair Encoding (BPE)

NLP tasks face an out-of-vocabulary problem which has largely been alleviated by BPE (Sennrich et al., 2016). BPE is a method that cuts words into sub-words based on statistical information from the training corpus.

There are three levels of texts in BPE: character level, sub-word level, and word level. BPE does not learn how to turn a word into a sub-word, but learns how to combine character-level units into sub-words. Firstly, BPE splits every word into separate characters, which are basic units. Then, BPE merges adjacent two basic units with the highest concurrent frequency. The combination of two units forms a new sub-word, which is also a basic

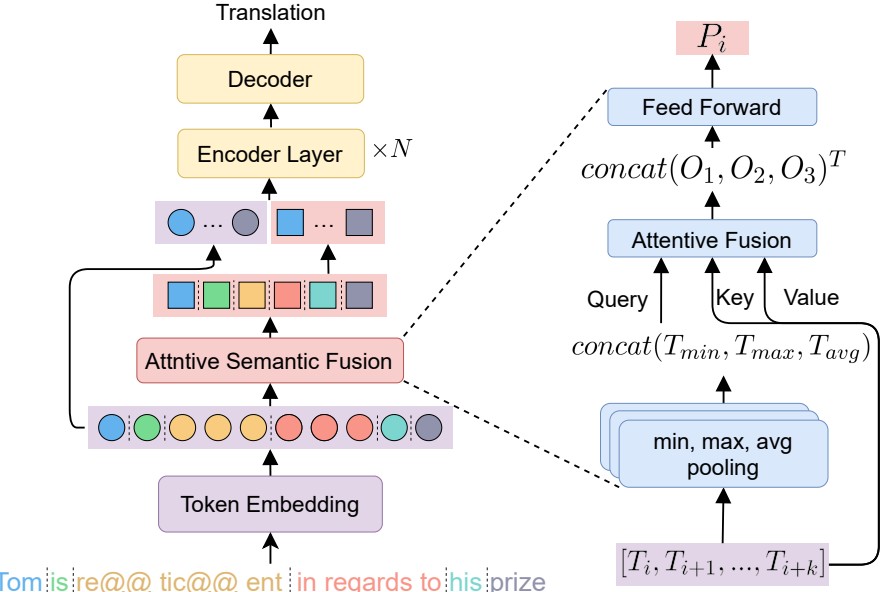

Figure 2: This picture illustrates an overview of SU4MT. The left part illustrates the whole model structure. We only modify the embedding layer of the encoder and keep others unchanged. The circles denote token-level representations and circles of the same color show they can form a semantic unit. The squares denote semantic-level representations. The right part illustrates the detailed structure of the Attentive Semantic Fusion (ASF) layer. The input is contiguous tokens that constitute a semantic unit and the output of it is a single semantic representation.

unit and it joins the next merge iteration as other units do. Merge operation usually iterates 10k-40k times and a BPE code table is used to record this process. These operations are called BPE learning, which is conducted on the training set only.

Applying BPE is to follow the BPE code table and conduct merge operations on training, validation, and test sets. Finally, a special sign "@@" is added to sub-words that do not end a word. For example, the word "training" may become two sub-words "train@@" and "ing".

## 3 Method

In this section, we present our methods in detail. As depicted in Figure 2, the proposed method introduces an additional Attentive Semantic Fusion (ASF) module between the Encoder Layers and the Token Embedding layer of the conventional Transformer model, and only a small number of extra parameters are added. The ASF takes the representations of several tokens that form a semantic unit as input, and it integrates the information from each token to output a single vector representing the united semantic representation. The sentence representations of both token-level and semantic-unit-level are then concatenated as the input of the first one of the encoder layers. Note that the

token-level representation is provided to the encoder layers to supplement detailed information. We also propose Word Pair Encoding (WPE), an offline method to effectively extract phrase spans that indicate the boundaries of semantic units.

### 3.1 Learning Semantic Unit Representations

In our proposed method, a semantic unit refers to a contiguous sequence of tokens that collectively form a unified meaning. This means not only phrases but also words composed of subwords can be considered as semantic units.

We extract token representations' features through pooling operations. Then, we utilize the attention mechanism to integrate the semantics of the entire semantic unit and map it to a fixed-length representation.

The semantic unit representations are obtained through the Attentive Semantic Fusion (ASF) layer, illustrated on the right side of Figure 2. It takes a series of token representations $T_{i \sim i+k} = [T_i, T_{i+1}, ..., T_{i+k}]$ as input, which can form a phrase of length $k + 1$. To constrain the output size as a fixed number, we leverage a characteristic of attention mechanism (Bahdanau et al., 2015a) that the size of attention output is determined by its query vector. Inspired by Xu et al. (2020), who

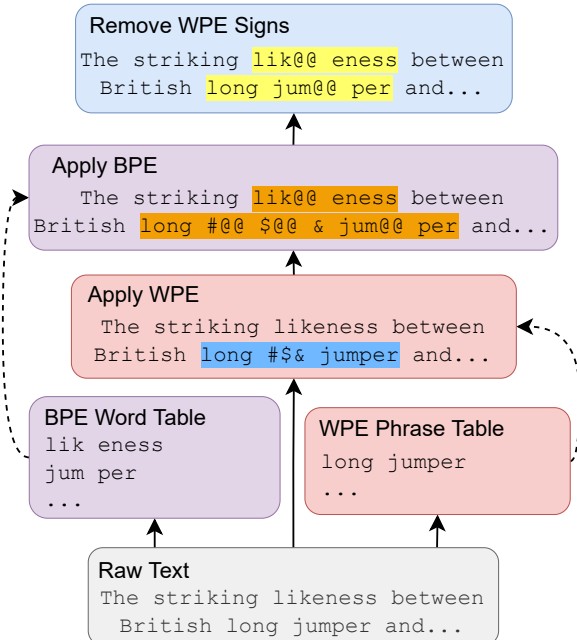

Figure 3: An example of how to get the span positions of semantic units in a subword-level sentence by applying WPE. The boxes are colored differently to distinguish different processing stages. Tokens that form a semantic unit are highlighted with a background color. The two highlighted strings in the uppermost box are the extracted semantic units.

have demonstrated max pooling and average pooling are beneficial features for learning phrase representations, we propose to leverage the concatenation of min pooling, max pooling, and average pooling to preserve more original information, yielding $T_{pool} = concat(T_{min}, T_{max}, T_{avg})$, which serves as the query vector for the attentive fusion layer. The attentive fusion layer uses $T_{i \sim i+k}$ as key and value vector and outputs the fused representation of length 3: $O = concat(O_1, O_2, O_3)$. The output is then transposed to be a vector with triple-sized $embed\_dim$ but a length of 1. This is followed by a downsampling feed-forward layer to acquire the single-token-sized semantic representation $P_i$.

In practical applications, we also consider the semantic units that are composed of only one token as phrases: a generalized form of phrases with a length of 1. Therefore, all tokens are processed by ASF to be transformed into semantic representations. By doing so, representations of all semantic units are in the same vector space.

## 3.2 Integrating Semantic Units

The left part of Figure 2 illustrates how the whole model works. The tokens in the input sentence

are printed in different colors, and the tokens of the same color can form a semantic unit. When SU4MT takes a sentence as input, the token embedding[1] maps these tokens into token-level representations, denoted by colored circles in Figure 2.

One copy of the token-level sentence representation is directly prepared to be a half of the next layer's input after being added with positional encoding. The other copy goes through the ASF layer and is transformed into the semantic-unit-level sentence representation, denoted by colored squares in Figure 2. It is then added with positional encoding, the position count starting from zero. Subsequently, the two levels of sentence representations are concatenated as the complete input of encoder layers. In practice, a normalization layer is applied to enhance the model stability.

Since the two levels of sentence representations are respectively added with positional encodings, they essentially represent two separate and complete sentences. This is like generating translations based on two different perspectives of a same natural language sentence, allowing the model to capture either the overall information or the fine-grained details as needed. To ensure the validity of concatenating the two levels of sentence representations, we conducted preliminary experiments. When a duplicate of the source sentence is concatenated to the original one as input of the conventional Transformer model, the translation quality does not deteriorate. This ensures the rationality of concatenating the two levels of sentence representations.

## 3.3 Extracting Semantic Spans

The semantic units encompass both words comprised of subwords and phrases. The former can be easily identified through the BPE mark "@@". However, identifying phrases from sentences requires more effort. Previous research often relied on parsing tools to extract phrases, which can be time-consuming. In our model, the syntactic structure information is not needed, and we only have to concern about identifying phrases. Therefore, we propose Word Pair Encoding (WPE) as a fast method for phrase extraction. Since our model only utilizes the phrase boundary information as an auxiliary guidance, it does not require extremely high accuracy. As a result, WPE strikes a good balance

---

[1]Token embedding is the same as word embedding in Transformer. We use this to distinguish tokens and words.

| Method | Base | | Big | |
|---|---|---|---|---|
| | **Param.** | **BLEU** | **Param.** | **BLEU** |
| Transformer* (Vaswani et al., 2017) | 65M | 27.3 | 213M | 28.4 |
| MG-SA* (Hao et al., 2019)† | 89.9M | 28.28 | 271.5M | 29.01 |
| LD* (Xu et al., 2020) | 173.0M | 28.67 | - | 29.60 |
| Proto-TF* (Yin et al., 2022)† | - | 28.49 | - | - |
| Transformer (Vaswani et al., 2017) | 63.40M | $28.46_{(27.29)}$ | 209.90M | $29.56_{(28.59)}$ |
| UMST (Li et al., 2022a)† | 68.65M | $29.24_{(28.19)}$ | 237.98M | $30.70_{(29.56)}$ |
| SU4MT | 66.69M | $\mathbf{29.80}^{\uparrow}_{(\mathbf{28.58})}$ | 232.99M | $\mathbf{30.89}^{\uparrow}_{(\mathbf{29.71})}$ |

Table 1: This table shows experimental results on En→De translation task. The proposed SU4MT approach is compared with related works that involve phrases or spans. The * sign denotes the results come from the original papers, and the † sign denotes the approaches use external parsing tools. Besides Multi-BLEU, we report SacreBLEU in brackets. The results show that our approach outperforms other methods w/ or w/o prior syntactic knowledge. The ↑ sign denotes our approach significantly surpasses Transformer ($p < 0.01$) and UMST ($p < 0.05$).

between quality and speed, making it suitable for the following stages of our approach.

Inspired by byte pair encoding (BPE) (Sennrich et al., 2016), we propose a similar but higher-level algorithm, word pair encoding (WPE), to extract phrases. There are two major differences between WPE and BPE: the basic unit and the merge rule.

The basic units in BPE are characters and pre-merged sub-words as discussed in section 2.2. Analogously, basic units in WPE are words and pre-merged sub-phrases. Besides, the merge operation does not paste basic units directly but adds a special sign "#$&" to identify word boundaries.

The merge rule means how to determine which two basic units should be merged in the next step. In BPE, the criterion is co-current frequency. In WPE, however, the most frequent co-current word pairs mostly consist of punctuations and stopwords. To effectively identify phrases, we change the criterion into a score function as shown in equation (3).

$$score = \frac{count(w1, w2) - \delta}{\sqrt{count(w1) \times count(w2)}} \quad (3)$$

$w_1$ and $w_2$ represent two adjacent basic units, and $\delta$ is a controllable threshold to filter out noises.

Notably, our WPE is orthogonal to BPE and we provide a strategy to apply both of them, as illustrated in Figure 3. Firstly, BPE word table and WPE phrase table are learned separately on raw text $X$. Then, WPE is applied to yield $X_W$, denoted by the red boxes. Next, we apply BPE to $X_W$ and obtain $X_{WB}$, denoted by the purple boxes. Finally, the special WPE signs are moved out and result in sub-word level sentences $X_B$, which is the same as applying BPE on raw text. The important thing

is we extract semantic units no matter whether the integrant parts are words or subwords. To sum up, WPE is applied to extract the position of semantic units, without changing the text.

In practice, there are some frequent long segments in the training corpus, bringing non-negligible noise to WPE. So we clip phrases with more than 6 tokens to avoid this problem.

## 4 Experiments

In this section, we conduct experiments on three datasets of different scales to validate the effectiveness and generality of the SU4MT approach.

### 4.1 Data

To explore model performance on small, middle, and large-size training corpus, we test SU4MT and related systems on WMT14 English-to-German (En→De), WMT16 English-to-Romanian (En→Ro), and WMT17 English-to-Chinese (En→Zh) translation tasks.

**Datasets**

For the En→De task, we use the script "*prepare-wmt14en2de.sh*" provided by Fairseq (Ott et al., 2019) to download all the data. The training corpus consists of approximately 4.5M sentence pairs. We remove noisy data from the training corpus by filtering out (1) sentences with lengths less than 3 or more than 250, (2) sentences with words composed of more than 25 characters, (3) sentence pairs whose ratio of source sentence length over target sentence length is larger than 2 or less than 0.5. After cleaning, the training corpus contains

| Method | Base | | Big | |
|---|---|---|---|---|
| | Param. | BLEU | Param. | BLEU |
| Transformer (Vaswani et al., 2017) | 60.92M | $34.42_{(34.23)}$ | 209.9M | $34.34_{(34.17)}$ |
| UMST (Li et al., 2022a)[†] | 60.83M | $34.66^{\uparrow}_{(34.50)}$ | 222.32M | $34.53_{(34.31)}$ |
| SU4MT | 66.69M | $\mathbf{34.87_{(34.71)}}^{\Uparrow}$ | 237.20M | $\mathbf{34.73_{(34.53)}}^{\Uparrow}$ |

Table 2: Experimental results on small-sized WMT16 En→Ro dataset. The [†] sign denotes the approach uses an external parsing tool. The $\Uparrow$ sign and $\uparrow$ denote the approaches significantly surpass the Transformer system, measured by SacreBLEU($p < 0.05$ and $p < 0.1$ respectively).

4.02M sentence pairs. We use *newstest2013* as the validation set and use *newstest2014* as the test set.

For the En→Zh task, we collect the corpora provided by WMT17 (Bojar et al., 2017). The training corpus consists of approximately 25.1M sentence pairs, with a notable number of noisy data. Therefore, we clean the corpus by filtering out non-print characters and duplicated or null lines, leaving 20.1M sentence pairs. We use *newsdev2017* and *newstest2017* as validation and test sets.

For the En→Ro task, the training corpus consists of 610K sentence pairs, and *newsdev2016* and *newstest2016* are served as validation and test sets.

**Data Preparation**

We use *mosesdecoder* (Koehn et al., 2007) for the tokenization of English, German, and Romanian texts, while the *jieba* tokenizer[2] is employed for Chinese texts. For the En-De and En-Ro datasets, we apply joint BPE with 32,768 merges and use shared vocabulary with a vocabulary size of 32,768. As for the En-Zh dataset, since English and Chinese cannot share the same vocabulary, we perform 32,000 BPE merges separately on texts in both languages. The vocabulary sizes are not limited to a certain number, and they are about 34K for English and about 52K for Chinese.

In our approach, the proposed WPE method is used to extract the boundaries of semantic units. For all three tasks, we employ the same settings: the $\delta$ value of 100 and 10,000 WPE merges. Finally, the resulting spans longer than 6 are removed.

**4.2 Implementation Details**

**Training Stage**

In SU4MT and the baseline method, the hyperparameters related to the training data scale and learning strategies remain unchanged across all tasks and model scales. We set the learning rate to $7e-4$

[2] https://github.com/fxsjy/jieba

and batch size to $32k$. Adam optimizer (Kingma and Ba, 2015) is applied with $\beta = (0.9, 0.98)$ and $\epsilon = 1e-8$. The dropout rate is adjusted according to the scales of model parameters and training data. For base-setting models, we set $dropout = 0.1$. For large-setting models, we set $dropout = 0.3$ for En→Ro and En→De tasks, and set $dropout = 0.1$ for En→Zh task.

For SU4MT, a pretrain-finetune strategy is applied for training stability. Specifically, we initialize our method by training a Transformer model for approximately half of the total convergence steps. At this point, the token embedding parameters have reached a relatively stable state, which facilitates the convergence of the ASF layer. The En→Ro task, however, is an exception. We train SU4MT models from scratch because it only takes a few steps for them to converge.

We reproduce UMST (Li et al., 2022a) according to the settings described in the paper. Note that the authors have corrected that they applied 20k BPE merge operations in the WMT16 En→Ro task, and we follow this setting in our re-implementation.

**Inference Stage**

For all experiments, we average the last 5 checkpoints as the final model and inference with it. Note that, we save model checkpoints for every epoch for En→Ro and En→De tasks and every 5000 steps for En→Zh task.

**Evaluation Stage**

We report the results with three statistical metrics and two model-based metrics. Due to space limitation, we exhibit two mainstream metrics here and display the complete evaluation in Appendix A. For En→Ro and En→De tasks, we report Multi-BLEU for comparison with previous works and Sacre-BLEU as a more objective and fair comparison for future works. For the En→Zh task, we discard Multi-BLEU because it is sensitive to the word

| Method | SacreBLEU | ChrF | Param. |
|---|---|---|---|
| *Base Setting* | | | |
| Transformer | 35.17 | 31.18 | 88.49M |
| SU4MT | **35.54** | **31.40** | 94.27M |
| *Big Setting* | | | |
| Transformer | 35.73 | 31.77 | 265.07M |
| SU4MT | **36.32** | **32.12** | 288.15M |

Table 3: Experimental results on En→Zh translation task are reported with SacreBLEU and ChrF. This is because the tokenization of Chinese is highly dependent on the segmentation tool, leading to the unreliability of Multi-BLEU as a metric.

| Encoder Input | SacreBLEU | Multi-BLEU |
|---|---|---|
| token+semantic | **28.58** | **29.80** |
| semantic×2 | 28.28 | 29.45 |
| token×2 | 28.14 | 29.29 |
| semantic only | 28.35 | 29.51 |
| token only | 27.54 | 28.64 |

Table 4: In this set of experiments, encoder input is varied to discover the effectiveness of concatenating sentence representations of both levels. Experiments are conducted on En→De task and the strategy of identifying semantic units is WPE 10k + BPE.

segmentation of Chinese sentences. Instead, we report SacreBLEU and ChrF (Popović, 2015). For model-based metrics, COMET (Rei et al., 2022)[3] and BLEURT (Sellam et al., 2020; Pu et al., 2021)[4] are leveraged.

### 4.3 Main Results

We report the experiment results of 3 tasks in Table 1, 2, 3. Of all the comparing systems, MG-SH (Hao et al., 2019), Proto-TF (Yin et al., 2022) and UMST (Li et al., 2022a) uses external tools to obtain syntactic information, which is time-consuming. Notably, our approach and LD (Xu et al., 2020) can work without any syntactic parsing tools and yield prominent improvement too. Apart from Multi-BLEU, we also report SacreBLEU in brackets. Results in Table 1 demonstrate that our approach outperforms other systems in both base and big model settings and significantly surpasses the systems without extra knowledge.

To make the evaluation of translation performance more convincing, we report detailed evaluation scores on five metrics, which are displayed in Appendix A.

## 5 Analysis

### 5.1 Ablation Study

To further understand the impact of providing different forms of source sentence representations to the encoder layer on model performance, we conduct three sets of ablation experiments based on the default settings of ASF, as shown in Table 4. We replace one of the two perspectives of sentence

representations with only one type, but duplicate it and concatenate it to the original sentence to eliminate the influence of doubled sentence length. As demonstrated by the experimental results, using only one perspective of sentence representation leads to a decrease in translation quality, indicating that diverse sentence representations complement each other and provide more comprehensive semantic information.

Furthermore, we remove the duplicated portion to investigate the impact of adding redundant information on translation. Surprisingly, when doubling the token-level sentence representation, SacreBLEU increases by 0.6, but when doubling the semantic-unit-level sentence representation, SacreBLEU even exhibits a slight decline. We speculate that repeating the information-sparse token-level sentence representation allows the model to learn detailed semantic information and to some extent learn overall semantic representation separately from the two identical representations. However, semantic-unit-level sentence representation is semantically dense. It's already easy for the model to understand the sentence and duplicating it may introduce some noises. In conclusion, using both perspectives of semantic representation together as input maximizes the provision of clear, comprehensive semantic and detailed linguistic features.

### 5.2 Influence of Granularity

We also discuss the criterion of how to consider a phrase as a semantic unit. We change the WPE merge steps to control the granularities. In Table 5, "BPE" indicates we treat all bpe-split subwords as semantic units. "RANDOM" means we randomly select consecutive tokens as semantic units until the ratio of these tokens is the same as that of our default setting (approximately 36%). The results

[3] https://huggingface.co/Unbabel/wmt22-comet-da

[4] https://storage.googleapis.com/bleurt-oss-21/BLEURT-20.zip

| Granularity | SacreBLEU | Multi-BLEU |
|---|---|---|
| WPE  5k +BPE | 34.62 | 34.71 |
| WPE 10k+BPE | **34.71** | **34.87** |
| WPE 15k+BPE | 34.54 | 34.60 |
| WPE 10k | 34.32 | 34.45 |
| BPE | 34.31 | 34.46 |
| RANDOM | 34.35 | 34.48 |

Table 5: In the table, we regard different contiguous tokens as semantic units to demonstrate the effectiveness of our method in extracting and leveraging semantic units. Experiments are conducted on En→Ro task. "5k", "10k", and "15k" denote the number of WPE merges, "BPE" indicates subwords that form a word are treated as semantic units.

demonstrate it is essential to integrate semantic meanings of subwords.

### 5.3  Performance on Different Span Number

To explore the effectiveness of our method in modeling semantic units, we divide the test set of the En→De task into five groups based on the number of semantic-unit spans contained in the source sentences (extracted using WPE 10k+BPE). We calculate the average translation quality of our method and the comparison systems in each group. The results are presented in Figure 4, where the bar graph represents the proportion of sentences in each group to the total number of sentences in the test set, and the line graph shows the variation in average SacreBLEU. When the span count is 0, SU4MT degrades to the scenario of doubling the token-level sentence representation, thus resulting in only a marginal improvement compared to the baseline. However, when there are spans present in the sentences, SU4MT significantly outperforms the baseline and consistently surpasses the comparison system (Li et al., 2022a) in almost all groups. This demonstrates that SU4MT effectively models semantic units and leverages them to improve translation quality.

### 5.4  Effectiveness of Modeling Semantic Units

To evaluate the efficacy of the proposed approach in modeling the integral semantics of semantic units, we calculate the translation accuracy of target-side words that correspond to the source-side semantic units. For this purpose, we leverage the *Gold Alignment for German-English Corpus* dataset[5]

---
[5] https://www-i6.informatik.rwth-aachen.de/goldAlignment/index.php

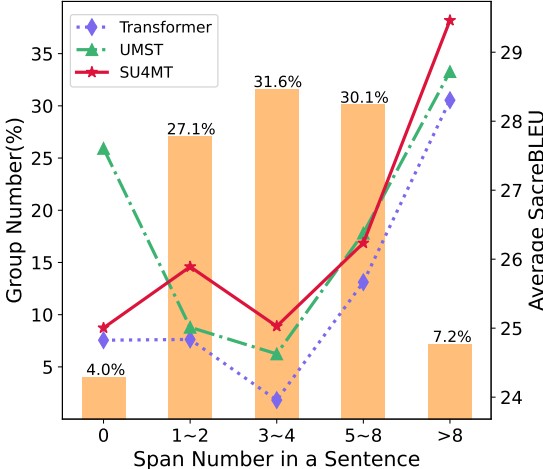

Figure 4: En→De test set is divided into 5 groups by the number of spans within a sentence. Lines exhibit the average SacreBLEU scores of sentences among each group. Bars show the number of sentences in each group. In most scenarios, SU4MT significantly outperforms the contrast methods.

proposed by RWTH (Vilar et al., 2006), which consists of 508 sentence pairs and human-annotated word alignments. The alignments are labeled either S (sure) or P (possible). We train SU4MT and baseline systems on WMT14 En-De dataset and evaluate them on the RWTH alignment dataset. The recall rate of target-side words that correspond to source-side semantic units is reported as the metric, i.e., the proportion of correctly translated semantic units. The results are presented in Table 6.

| Method | recall(S) | recall(P&S) | SacreBLEU |
|---|---|---|---|
| Transformer | 68.14 | 65.99 | 25.27 |
| UMST | 69.18 | 66.96 | 25.96 |
| SU4MT | **69.29** | **67.24** | **26.58** |

Table 6: Our approach presents more accurate translation on words that align with semantic units. In "recall(S)", only the words with "sure" alignment are calculated, while in "recall(P&S)", we calculate words with both "sure" and "possible" alignment.

## 6  Related Works

### 6.1  Multiword Expression

Multiword Expression (MWE) is a notion similar to semantic units that proposed in our work. Several works have proposed definitions of MWE (Carpuat and Diab, 2010; Calzolari et al., 2002; Sag et al., 2002; Baldwin and Kim, 2010). Although MWE

was a hot research topic, leveraging it in neural machine translation is hindered by its intrinsic characteristics like discontinuity (Constant et al., 2017), e.g., the expression "as ... as ...".

Despite the large overlap between these two notions, we claim the distinctions of semantic units. For empirical usage, we require semantic units to be strictly consecutive tokens. Subword tokens are also considered a part of a semantic unit.

### 6.2 Extraction of Phrases

From the perspective of extracting semantic phrases, some of these methods require an additional parsing tool to extract semantic units and their syntactic relations, which may cost days in the data preprocessing stage. Phrase extraction has been explored by various methods, such as the rule-based method (Ahonen et al., 1998), the alignment-based method (Venugopal et al., 2003; Vogel, 2005; Neubig et al., 2011), the syntax-based method (Eriguchi et al., 2016; Bastings et al., 2017), and the neural network-based method (Mingote et al., 2019).

### 6.3 Translation with Phrases

Several works have discussed leveraging semantic or phrasal information for neural machine translation. Some of them take advantage of the span information of phrases but do not model phrase representation. This kind of method modifies attention matrices, either conducting matrix transformation to integrate semantic units (Li et al., 2022a) or applying attention masks to enhance attention inside a semantic span (Slobodkin et al., 2022). Other works model a united representation of semantic units. Xu et al. (2020) effectively learns phrase representations and achieves great improvement over the baseline system. However, a complicated model structure is designed to leverage semantic information, resulting in heavy extra parameters. MG-SA (Hao et al., 2019) models phrase representations but requires them to contain syntactic information instead of semantics, which narrows the effectiveness of modeling phrases. Proto-TF (Yin et al., 2022) uses semantic categorization to enhance the semantic representation of tokens to obtain a more accurate sentence representation, but it fails to extend the method into phrasal semantic representations.

## 7 Conclusions

In this work, we introduce the concept of semantic units and effectively utilize them to enhance neural machine translation. Firstly, we propose Word Pair Encoding (WPE), a phrase extraction method based on the relative co-occurrence of words, to efficiently extract phrase-level semantic units. Next, we propose Attentive Semantic Fusion (ASF), a phrase information fusion method based on attention mechanism. Finally, we employ a simple but effective approach to leverage both token-level and semantic-unit-level sentence representations. Experimental results demonstrate our method significantly outperforms similar methods.

## Limitations

Our approach aims to model the semantic units within a sentence. However, there are still a few sentences that contain semantic units as Figure 4 shows. In this scenario, SU4MT brings less improvement in translation performance. Therefore, the advantage of our approach may be concealed in extreme test sets. Admittedly, SU4MT requires more FLOPs than Transformer does, which is an inevitable cost of leveraging extra information. There are also potential risks in our work. Limited by model performance, our model could generate unreliable translations, which may cause misunderstanding in cross-culture scenarios.

## Acknowledgements

We thank the anonymous reviewers and chairs for their insightful comments.

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

# A  More Evaluation Metrics

To better evaluate our model and baseline systems, and to provide a complete benchmark for future research on machine translation, we report the results on five metrics. "Multi-BLEU", "SacreBLEU", and "ChrF" are statistical metrics; "COMET" and "BLEURT" are model-based metrics.

| Method | Parameters | Multi-BLEU | SacreBLEU | ChrF | COMET | BLEURT |
|---|---|---|---|---|---|---|
| *Base setting* | | | | | | |
| Transformer | 65M | 28.46 | 27.29 | 57.68 | 83.11 | 71.74 |
| UMST | 68.65M | 29.24 | 28.19 | 57.75 | 83.20 | 71.75 |
| SU4MT | 66.69M | **29.80** | **28.58** | **57.84** | **83.28** | **72.05** |
| *Large setting* | | | | | | |
| Transformer | 213M | 28.56 | 28.59 | 57.25 | 83.76 | 72.63 |
| UMST | 237.98M | 30.70 | 29.56 | **58.76** | 84.25 | 73.32 |
| SU4MT | 232.99M | **30.89** | **29.71** | 58.69 | **84.44** | **73.61** |

Table 7: Experimental results on En→De translation task. This a complete-metric version of Table 1.

| Method | Parameters | Multi-BLEU | SacreBLEU | ChrF | COMET | BLEURT |
|---|---|---|---|---|---|---|
| *Base setting* | | | | | | |
| Transformer | 60.92M | 34.42 | 34.23 | 60.14 | 81.08 | 71.97 |
| UMST | 60.83M | 34.66 | 34.50 | 60.38 | 81.49 | 72.36 |
| SU4MT | 66.69M | **34.87** | **34.71** | **60.59** | **81.71** | **72.78** |
| *Large setting* | | | | | | |
| Transformer | 209.90M | 34.34 | 34.17 | 60.33 | 81.52 | 72.56 |
| UMST | 222.32M | 34.58 | 34.45 | **60.53** | **81.93** | **72.88** |
| SU4MT | 237.20M | **34.73** | **34.53** | 60.29 | 81.76 | 72.61 |

Table 8: Experimental results on En→Ro translation task. This a complete-metric version of Table 2.

| Method | Parameters | Multi-BLEU | SacreBLEU | ChrF | COMET | BLEURT |
|---|---|---|---|---|---|---|
| *Base setting* | | | | | | |
| Transformer | 88.49M | 21.40 | 35.17 | 31.18 | 82.96 | 63.24 |
| UMST | 121.64M | 21.37 | 35.52 | 31.33 | 83.04 | 63.26 |
| SU4MT | 94.27M | **21.51** | **35.54** | **31.40** | **83.29** | **63.31** |
| *Large setting* | | | | | | |
| Transformer | 265.07M | 22.12 | 35.73 | 31.77 | 83.53 | 63.71 |
| UMST | 343.94M | **22.29** | **36.54** | **32.31** | 83.68 | 64.21 |
| SU4MT | 288.15M | 21.96 | 36.32 | 32.12 | **83.78** | **64.31** |

Table 9: Experimental results on En→Zh translation task. This a complete-metric version of Table 3.

## B   Scaling up Training corpus

We apply our approach to an even larger dataset: WMT14 English-to-French(En→Fr), the training corpus of which consists of approximately 35M sentence pairs. We apply joint BPE with 40,000 merges and use shared vocabulary with size 40K. The hyper-parameters of WPE are $\delta = 100$ and 10,000 merges. We save model checkpoints for every 5,000 steps and average the last 5 checkpoints for inference.

| Method | Parameters | Multi-BLEU | SacreBLEU | ChrF | COMET | BLEURT |
|---|---|---|---|---|---|---|
| *Base setting* | | | | | | |
| Transformer | 64.62M | 40.60 | 37.38 | 63.62 | 84.03 | 70.00 |
| SU4MT | 70.39M | **41.09** | **37.82** | **63.98** | **84.49** | **70.74** |
| *Large setting* | | | | | | |
| Transformer | 217.32M | 41.46 | 38.29 | 64.27 | 85.09 | 71.43 |
| SU4MT | 240.40M | **42.00** | **38.90** | **64.69** | **85.54** | **72.29** |

Table 10: Experimental results on En→Fr translation task.

## C   English-Targeted Experiment

Besides experimenting with English as the source language, we further implement our approach on English-targeted translation datasets. Specifically, we leverage the same datasets as mentioned in 4.1, but put English as the target side. Table11 shows the translation performance on five metrics.

| | Method | Multi-BLEU | SacreBLEU | ChrF | COMET | BLEURT |
|---|---|---|---|---|---|---|
| De-En | Transformer | 32.60 | 31.65 | 57.96 | 82.94 | 71.04 |
| | SU4MT | **33.07** | **32.02** | **58.31** | **83.21** | **71.37** |
| Ro-En | Transformer | 33.83 | 33.67 | 59.90 | 79.45 | 67.09 |
| | SU4MT | **34.44** | **34.21** | **60.28** | **80.00** | **67.66** |
| Zh-En | Transformer | 24.40 | 23.86 | 52.97 | 81.04 | 66.88 |
| | SU4MT | **24.97** | **24.41** | **53.40** | **81.22** | **67.03** |

Table 11: Results of English-targeted experiments, which are conducted on WMT14 De→En dataset, WMT16 Ro→En dataset, and WMT17 Zh→En dataset. All models are in base scale setting.