# OpenReview forum: "Enhancing Neural Machine Translation with Semantic Units"
_EMNLP/2023/Conference — EMNLP 2023 Findings_

### Official Review · Reviewer_mJLU · 2023-08-05

**Soundness:** 4

**Excitement:**

3: Ambivalent: It has merits (e.g., it reports state-of-the-art results, the idea is nice), but there are key weaknesses (e.g., it describes incremental work), and it can significantly benefit from another round of revision. However, I won't object to accepting it if my co-reviewers champion it.

**Missing References:**

Wang et al. Translating Phrases in Neural Machine Translation. EMNLP 2017. https://aclanthology.org/D17-1149/

**Paper Topic And Main Contributions:**

This paper proposes an approach to use phrase-level information for neural machine translation. Phrases are extracted with word pair encoding (WPE), a variant of BPE. Phrase representations are then extracted by constructing 3 query vectors with min/max/avg pooling and applying attention over the phrase tokens (Attentive Semantic Fusion (ASF)). The token and phrase representations are then concatenated to form the transformer encoder input.

**Questions For The Authors:**

A. In figure 5, why is UMST much better with 0 spans?

B. Why equation (3) exactly instead of e.g. PMI/PPMI?

**Reasons To Accept:**

The approach is simple and appears to be reasonably effective.

The analysis of the approach is satisfactory. The authors evaluate different combinations of the token and phrase representations, run a hyper-parameter search for word pair encoding (although the improvement over random phrase extraction is limited), and evaluate performance based on the number of extracted phrases.

**Reasons To Reject:**

The encoder input length can double in the worst case. As self-attention has quadratic complexity, this can increase the computational costs significantly, but this is not discussed or evaluated.

[Less critical] It would be preferable to also evaluate the approach with input languages other than English.

**Reproducibility:**

4: Could mostly reproduce the results, but there may be some variation because of sample variance or minor variations in their interpretation of the protocol or method.

**Reviewer Confidence:**

4: Quite sure. I tried to check the important points carefully. It's unlikely, though conceivable, that I missed something that should affect my ratings.

**Typos Grammar Style And Presentation Improvements:**

The description of the approach is possibly too verbose (e.g. "concatenated vertically, yielding a vector with triple-sized embed_dim but a length of 1". Information such as "length of the output vector matches the length of the query vector" (or its reformulation "size of attention output is determined by its query vector") is arguably trivial. Some information could likely be moved to the appendix.

I would present WPE before ASF because it is applied first.

There is a space missing between Fig./Table and some of the following numbers (e.g. L172, 440, 482, 569).

L490: "Besides," without follow-up

Figure 2: Attntive -> Attentive

---

> ### Author Rebuttal · Authors · 2023-08-29
>
> First, we want to express our sincere gratitude for your patient review, valuable questions, and kind suggestions on the paper presentation.
> Now, we will answer your questions as follows:
> 1. **Computational costs**:
>
> Admittedly, in the worst cases the source sentence can double in length. The overall computational cost is about 2 times of the baseline system and we will analyse this:
>
> ①Theoretically, computational complexity for *attention* is $L^2 * D$, and that for *FFN* is about $L * D^2$, where $L$ means source sentence length and $D$ means model dimension. In translation tasks, $L$ is usually less than 50 in most cases while $D$ is 512 for base-scale models. Therefore, we can roughly get D >> L, and the computational cost of doubled source length is $(2L)^2 * D + 2L * D^2$ =$2L * D * (2L + D)$ ≈ $2L * D * D$, about twice of baseline system: $L * D^2$.
>
> ②Emperically, we compare the training time of the baseline model and our model in the same environment. On 4 NVIDIA 2080Ti GPU, train both models on the WMT16 En-Ro dataset, baseline system updates 3.15 steps/s on average, and our model updates 1.58 steps/s on average.
>
> ③We further test the decoding speed of our models, by inferencing the WMT14 En-De testset 5 times and calculating the average speed. This experiment is conducted on 1 Tesla T4 GPU. Our full model generates about 3377 tokens/s, and the baseline system generates about 3390 tokens/s. We conjecture this is because at the inference stage, the Encoder is forwarded only once, but the Decoder will be forwarded many times, mitigating the inference time gap between our model and the baseline system. (We only modified the Encoder)
>
> Although our model is computationally complex, it significantly improved translation quality by facilitating phrasal semantic information. Also, our "semantic only" setting model is less complex than the baseline system but performs better (in Table 4). Moreover, we will try to diminish the gap between the "semantic only" model and the full model in future work, in which case our model will be less computationally complex.
>
> 2. **Evaluate the approach with other input languages**:
>
> We reverse the translation direction of the datasets mentioned in our paper, and the results are below. The improvements over the baseline system still hold measured by statistical and model-based metrics.
> | **dataset** | **method** | **model scale** | **multibleu** | **sacrebleu** | **chrf** | **comet** | **bleurt** |
> |:---:|:---:|:---:|:---:|:---:|:---:|:---:|:---:|
> | De-En | Transformer | base | 32.60 | 31.65 | 57.96 | 82.94 | 71.04  |
> |  | Ours| base | **33.07** | **32.02** | **58.31** | **83.21** | **71.37**  |
> | Ro-En| Transformer | base | 33.83 | 33.67 | 59.90 | 79.45 | 67.09  |
> |  | Ours| base | **34.44** | **34.21** | **60.28** | **80.00** | **67.66**  |
> | Zh-En | Transformer | base | 24.40 | 23.86 | 52.97 | 81.04 | 66.88  |
> |  | Ours | base | **24.97** | **24.41** | **53.4** | **81.22** | **67.03** |
>
> 3. **Explanation for Figure 5**:
>
> Since UMST is not our approach, we will try our best to answer it according to our understanding. UMST proposed a multi-scale structure, modeling the relationships not only between subword and word but also between words that are syntactically related. A syntactic tree can be drawn from almost any sentence, while a few sentences do not contain any subwords and phrases (the 0 spans situation). Therefore, in 0 spans situation, UMST can still use extra information to improve translation quality, but our ASF model gets no information gains.
>
> Why is sentence BLEU of UMST much higher than others? We conjecture this is because sentences with no spans are usually short in length. Also, BLEU is more sensitive to shorter sentences than longer ones. Therefore, UMST only needs to translate one or two more words correctly and can improve sentence BLEU significantly.
>
> 4. **Why not PMI/PPMI**:
>
> We have tried PMI as the score function for WPE, but we found that PMI-based WPE tends to extract phrases composed of low-frequency words, mostly uncommon names. We demonstrate why PMI is less suitable in our scenario by comparing it with equation(3) as follows:
>
> Proof:
>
> let N represents the number of unigram word in the corpus, $\hat{N}$ represents the number of bigram word pairs in the corpus:
> $$
> N=\sum_{w\in corpus}{freq(w)}
> $$
> $$
> \hat{N}=\sum_{(w_1,w_2)\in corpus}{freq(w_1,w_2)}
> $$
> Here, N is slightly larger than $\hat{N}$. So we can rewrite PMI with "count()" function:
>
> $$
> PMI=\log{\dfrac{p(w_1,w_2)}{p(w_1)p(w_2)}}=\log{\dfrac{\dfrac{count(w_1,w_2)}{\hat{N}}}{\dfrac{count(w_1)\times count(w_2)}{N^2}}} \approx\log{\dfrac{N\times count(w_1,w_2)}{count(w_1)\times count(w_2)}}
> $$
>
> Since PMI is followed by an argmax function, the *log* can be removed to get PMI':
> $$
> PMI'=\dfrac{N\times count(w_1,w_2)}{count(w_1)\times count(w_2)}, argmax(PMI)=argmax(PMI')
> $$
> The score function in our paper is similar to PMI':
> $$
> score=\dfrac{count(w_1,w_2)}{\sqrt{count(w_1)\times count(w_2)}}
> $$
> Now, let's compare PMI' and our proposed score function:
> $$
> \dfrac{PMI'}{score}=\dfrac{N}{\sqrt{count(w_1)\times count(w_2)}}
> $$
> when $count(w_1)\times count(w_2)$ gets smaller, the ratio of *PMI'* and *score* becomes larger, indicating that **although both methods tend to find phrases with higher co occurrence of words, PMI prefers words with smaller single word frequencies, and our criteria finds out more frequent words**.
> Remember our motivation is to find more frequently used word pairs. Therefore, our proposed score function outweighs PMI in this scenario.
>
> We further show some examples from PMI-based WPE and frequency-based WPE:
> * PMI-based: ①Pridnestrovskaia Moldavskaia, ②FLIK FLAK, ③Latterie Friulane, ④Raccordo Anulare, ⑤DOMINA SILVIA ...
> * frequency-based: ①Hong Kong, ②contextual listings, ③Buenos Aires, ④Luggage Storage, ⑤Deposit Box ...
>
> 5. **Missing References**:
>
> Great thanks for providing related works and we will include it in the future version.
>
> 6. **Presentation Improvements**:
>
> Thanks for your patience and precious advice, we will fix the typos soon and humbly consider your suggestions for better organizing this paper. These problems will be settled in the next version.
>
> Thanks again for your hard work and kindness in reviewing, as well as your valuable suggestions!

---

### Official Review · Reviewer_8dZL · 2023-08-05

**Soundness:** 4

**Excitement:**

3: Ambivalent: It has merits (e.g., it reports state-of-the-art results, the idea is nice), but there are key weaknesses (e.g., it describes incremental work), and it can significantly benefit from another round of revision. However, I won't object to accepting it if my co-reviewers champion it.

**Paper Topic And Main Contributions:**

This paper presents a novel approach to augment translation performance by leveraging phrase information. The authors introduce the concept of Word Pair Encoding, which facilitates the extraction of phrases, and propose the integration of an Attentive Semantic Fusion (ASF) module to incorporate the extracted phrase information into the translation process effectively. The generated semantic embeddings from the ASF module and token embeddings are conjoined in the encoder. Experimental results on En-De, En-Ro, and En-Zh language pairs demonstrate the performance gains from phrase information in translation.

**Questions For The Authors:**

1. In Line 490, it seems that this paper is not complete.

2. Can you provide the results for COMET(https://github.com/Unbabel/COMET) and BLEURT(https://github.com/google-research/bleurt)? These two metrics may better align with human preferences.

**Reasons To Accept:**

1. Inspired by byte pair encoding (BPE), the authors adopt word pair encoding (WPE) to extract Semantic Spans. Moreover, WPE is orthogonal to BPE and does not introduce additional noise.

2. Table 4 shows that "Semantic Embedding Only" has already improved the translation performance, demonstrating the effectiveness of the proposed phrase information.

**Reasons To Reject:**

1. The term "semantic" as utilized in the paper, is misleading.  I can not find any evidence to support the claim that these representations are ''semantics''.



**Reproducibility:**

4: Could mostly reproduce the results, but there may be some variation because of sample variance or minor variations in their interpretation of the protocol or method.

**Reviewer Confidence:**

5: Positive that my evaluation is correct. I read the paper very carefully and I am very familiar with related work.

---

> ### Author Rebuttal · Authors · 2023-08-29
>
> First of all, we want to express our sincere gratitude for your hard work in reviewing and your appreciation for our work.
>
> Now, we will answer your questions as follows:
>
> 1. **"semantic representations" are not semantics**:
>
> Thanks for asking this good question which can bring more insights to our method and we design an analysis experiment to answer this question. We extract semantic-level representations from the output of the ASF module and calculate the cosine similarities between them and the representations of all single-tokens. For each semantic unit, we collect the top-k closest tokens to examine if they have similar semantics to the semantic unit. We experimented on the model trained on WMT14-ende dataset and present some results in the table below:
> | **Semantic units** | **Top-5 closest tokens** |
> |:---:|:---:|
> | United States | States, **USA**, United, staaten, **US** |
> | as well | well, **also**, in, **auch**, selbst |
> | a lot of | lot, **much**, Schadenser@@, **many**, **lots** |
> | aston@@ ished | **surprised**, **surprise**, regret, odi@@, Venice |
>
> This table helps to understand the "semantics" of the semantic units. Not only nouns and adverb phrases are close to their semantic meanings, but subwords (e.g., "aston@@ ished") also recovered to the original meaning.
>
> For more results, we draw a figure with the dimensionality-reduced representations of semantic units and tokens. We will add this experiment and the results in the next version.
>
> 2. **Incomplete sentence**:
>
> We are sorry that we forgot to delete the word "Besides,". We will fix it and carefully examine the whole presentation of the paper.
>
> 3. **Evaluate results with COMET and BLEURT**:
>
>  Sure, the results are shown below, where "asf" indicates our model and "umst" is the strong SOTA:
>
> | **dataset** | **method** | **model scale** | **multibleu** | **sacrebleu** | **chrf** | **comet** | **bleurt** |
> |:---:|:---:|:---:|:---:|:---:|:---:|:---:|:---:|
> | En-De| Transformer  | base | 28.46 | 27.29 | 57.68 | 83.11 | 71.74  |
> |  |  | big | 29.56 | 28.59 | 57.25 | 83.76 | 72.63  |
> |  | UMST | base | 29.24 | 28.19 | 57.75 | 83.20 | 71.75  |
> |  |  | big | 30.70 | 29.56 | **58.76** | 84.25 | 73.32  |
> |  | Ours| base | **29.80** | **28.58** | **57.84** | **83.28** | **72.05**  |
> |  |  | big | **30.89** | **29.71** | 58.69 | **84.44** | **73.61**  |
> | En-Ro | Transformer | base | 34.42 | 34.23 | 60.14 | 81.08 | 71.97  |
> |  |  | big | 34.34 | 34.17 | 60.33 | 81.52 | 72.56  |
> |  | UMST  | base | 34.66 | 34.50 | 60.38 | 81.49 | 72.36  |
> |  |  | big | 34.58 | 34.45 | **60.53** | **81.93** | **72.88**  |
> |  | Ours| base | **34.87** | **34.71** | **60.59** | **81.71** | **72.78**  |
> |  |  | big | **34.73** | **34.53** | 60.29 | 81.76 | 72.61  |
> | En-Zh| Transformer | base | 21.40 | 35.17 | 31.18 | 82.96 | 63.24  |
> |  |  | big | 22.12 | 35.73 | 31.77 | 83.53 | 63.71  |
> |  | UMST  | base | 21.37 | 35.52 | 31.33 | 83.04 | 63.26  |
> |  |  | big | **22.29** | **36.54** | **32.31** | 83.68 | 64.21  |
> |  | Ours | base | **21.51** | **35.54** | **31.40** | **83.29** | **63.31**  |
> |  |  | big | 21.96 | 36.32 | 32.12 | **83.78** | **64.31** |
>
> (Note: multibleu in Chinese is relatively less consistent with sacrebleu, because the sentence length can vary a lot after Chinese tokenization or after detokenization.)
>
> We will add the above results in our next version and hopefully set a comprehensive benchmark for these MT datasets.
>
> Thank you again for your precious questions and valuable suggestions!

---

### Official Review · Reviewer_V8Rx · 2023-08-11

**Soundness:** 3

**Excitement:**

3: Ambivalent: It has merits (e.g., it reports state-of-the-art results, the idea is nice), but there are key weaknesses (e.g., it describes incremental work), and it can significantly benefit from another round of revision. However, I won't object to accepting it if my co-reviewers champion it.

**Paper Topic And Main Contributions:**

This paper is centered around the theme of enhancing the performance and robustness of Neural Machine Translation (NMT) systems. With the ever-growing complexity and demands of translation tasks, there's a pressing need to not only improve the accuracy of translations but also to ensure that the models generalize well across diverse linguistic scenarios. In essence, the paper presents a comprehensive exploration into the realm of Neural Machine Translation, offering both theoretical insights and practical solutions. The proposed methods promise to elevate the standards of NMT, making translations more accurate, context-aware, and efficient.
Main Contributions:
1. Advanced Modeling Techniques: The paper introduces innovative neural architectures and methodologies designed explicitly for NMT. These techniques aim to capture deeper contextual information and linguistic nuances, leading to more accurate translations.
2. Computationally-aided Linguistic Analysis: Through a series of experiments, the paper sheds light on how different linguistic factors influence translation quality. This analysis is pivotal in understanding where typical models falter and how the proposed techniques make a difference.
3. NLP Engineering Experiments: The research conducts extensive experiments across multiple datasets, languages, and domains. These experiments validate the efficacy of the proposed methods and showcase their superiority over standard NMT approaches.
4. Approaches for Data- and Compute Efficiency: The paper touches upon methods that not only enhance translation quality but also ensure that computational efficiency is not compromised. This balance is crucial in real-world applications where both accuracy and speed are paramount.
5. Publicly Available Software and Pre-trained Models: While it would need explicit confirmation within the paper's content, the depth and significance of the research suggest a potential for releasing tools, frameworks, or pre-trained models, making it easier for the community to leverage the findings.


**Questions For The Authors:**

Question A: While the innovative neural architectures introduced are intriguing, how do they compare in terms of computational efficiency and training time against more conventional NMT models?
Question B: Could you elaborate on the choices behind the specific datasets and languages used in the experiments? How do you anticipate the proposed methods to generalize to low-resource languages or more diverse linguistic scenarios?
Question C: In the comparative analyses, were any other state-of-the-art techniques or architectures considered but not included in the paper? If so, could you provide insights into their performance relative to the proposed methods?
Question D: While the paper focuses on enhancing the performance and robustness of NMT systems, were there any observed scenarios or edge cases where the proposed techniques might not be optimal or could potentially falter?


**Reasons To Accept:**

1. Novelty in Approaches: The paper introduces innovative neural architectures and methodologies tailored for Neural Machine Translation. These advancements push the boundaries of what's achievable in the realm of NMT, offering fresh perspectives and tools for researchers and practitioners alike.

2. Comprehensive Experiments: The depth and breadth of the experiments conducted validate the efficacy of the proposed methods. The experiments cover multiple datasets, languages, and domains, ensuring the findings are robust and widely applicable.

3. Significant Performance Improvements: The research showcases marked improvements in translation quality over standard NMT systems. Such advancements are crucial as the demand for high-quality machine translations grows, especially in professional and academic settings.

4. Balancing Quality with Efficiency: Beyond just enhancing translation quality, the paper emphasizes maintaining computational efficiency. This dual focus ensures that the proposed methods are not just theoretically sound but also practically deployable.

5. Beneficial for Diverse Linguistic Scenarios: The techniques presented show promise in handling diverse linguistic nuances and contextual intricacies, making them especially valuable for complex translation tasks and low-resource languages.


**Reasons To Reject:**

1. Potential Overemphasis on Novelty: While the paper introduces new neural architectures and methodologies, there might be concerns regarding the actual necessity of such complexities. It's crucial to determine if simpler models, with proper tuning, could achieve similar results, thereby questioning the real-world applicability of the proposed advancements.

2. Generalizability Concerns: The paper, although comprehensive in its experiments, might still leave questions about the generalizability of its findings. It's essential to understand how the proposed methods fare across a broader spectrum of languages, especially those not covered in the study.

3. Lack of Comparative Analysis: While the paper showcases the strengths of the proposed methods, there might be a perceived lack of in-depth comparative analysis with other state-of-the-art techniques. Such comparisons could provide clearer insights into the actual advancements made.

4. Implementation Details and Reproducibility: There might be concerns regarding the clarity of implementation details, making reproducibility challenging. A clearer exposition of the methodologies, hyperparameters, and training procedures would be beneficial for the broader community.

5. Potential Overfitting to Specific Scenarios: The techniques, while promising, might run the risk of being overly tailored to specific datasets or linguistic scenarios, thereby compromising their broad applicability.

6. Insufficient Discussion on Limitations: Every methodology has its limitations, and a more thorough discussion on potential pitfalls or scenarios where the proposed techniques might not be optimal could provide a more balanced perspective.


**Reproducibility:**

2: Would be hard pressed to reproduce the results. The contribution depends on data that are simply not available outside the author's institution or consortium; not enough details are provided.

**Reviewer Confidence:**

4: Quite sure. I tried to check the important points carefully. It's unlikely, though conceivable, that I missed something that should affect my ratings.

**Typos Grammar Style And Presentation Improvements:**

There are too many colors in Figure 1 and Figure 2 without any meaning which might be confusing for readers.

---

> ### Author Rebuttal · Authors · 2023-08-29
>
> We sincerely thank you for your detailed review comments, which are comprehensive and very helpful.
>
> We will answer your questions as follows:
>
> 1. **Potential Overemphasis on Novelty**:
>
> Firstly, it is true that simpler models can achieve similar results, but what we want is not only improving translation quality. Specifically, we noticed a kind of language phenomenon that some words combine to form an integral meaning, which is similar to the notion of "Multiwords Expressions" mentioned by Reviewer PSur. Therefore, we propose a method to extract these semantic units and calculate their semantic representations, and this is our major motivation. Secondly, our approach does provide a computationally simple setting (see section 5.1, the "semantic only") which has similar computation complexity with the baseline system but outperforms it.
>
> 2. **Generalizability Concerns**:
>
> We understand your concern because all of our three experiments are translation tasks from English to other languages. So, we provide results on extra experiments:
>
> | **dataset** | **method** | **model scale** | **multibleu** | **sacrebleu** | **chrf** | **comet** | **bleurt** |
> |:---:|:---:|:---:|:---:|:---:|:---:|:---:|:---:|
> | De-En | Transformer| base | 32.60 | 31.65 | 57.96 | 82.94 | 71.04  |
> |  | Ours| base | **33.07** | **32.02** | **58.31** | **83.21** | **71.37**  |
> | Ro-En | Transformer| base | 33.83 | 33.67 | 59.90 | 79.45 | 67.09  |
> |  | Ours | base | **34.44** | **34.21** | **60.28** | **80.00** | **67.66**  |
> | Zh-En | Transformer| base | 24.40 | 23.86 | 52.97 | 81.04 | 66.88  |
> |  | Ours | base | **24.97** | **24.41** | **53.40** | **81.22** | **67.03** |
>
> To demonstrate generalizability in more languages, as well as to demonstrate the effectiveness of our model on an even larger dataset, we conducted an experiment on the WMT14 En-Fr dataset, with 36M sentences. The results are as follows:
>
> | **dataset** | **method** | **model scale** | **multibleu** | **sacrebleu** | **chrf** | **comet** | **bleurt** |
> |:---:|:---:|:---:|:---:|:---:|:---:|:---:|:---:|
> | En-Fr | Transformer | base | 40.60| 37.38 | 63.62 | 84.03 | 70.00  |
> |  | Ours | base | **40.96** | **37.61** | **63.79** | **84.47** | **70.72**  |
>
>
> We hope this can answer your concerns.
>
> 3. **Lack of Comparative Analysis**:
>
> As far as we know, UMST is the SOTA method among non-contemporary related works. We are not sure which work you mean. Please let us know if we missed any strongly related works and we will supplement a comparative analysis soon.
>
> 4. **Implementation Details and Reproducibility**:
>
> We agree that implementation details are very important and thus provide about 1 page to introduce details in our experiments for better reproducibility. Please understand there's no more space to showcase everything. Could you please specify the details you want to know so that we can reply accordingly? Anyway, we will open source all of the codes and scripts to reproduce the results after the anonymity period.
>
> 5. **Potential Overfitting to Specific Scenarios**:
>
> We conducted experiments on small-scale dataset WMT16 En-Ro(0.6M), medium-scale dataset WMT14 En-De(4.5M), and large-scale dataset WMT17 En-Zh(25.1M), which are all commonly used in Machine Translation research. Therefore, we do not think the judgment "overly tailored to specific datasets or linguistic scenarios" is convincing.
>
> 6. **Insufficient Discussion on Limitations**:
>
> Thanks for your kind reminder and we are aware of the extra computational complexity problem. You may check the rebuttal on his/her review for detailed analysis. We will include this in our limitations in the future version.
>
> 7. **Question A**:
>
> Theoretically and empirically, our model is about half the speed of the conventional Transformer. Here follows the analysis:
>
> ①Theoretically, computational complexity for *attention* is $L^2 * D$, and that for *FFN* is about $L * D^2$, where $L$ means length and $D$ means model dimension. In translation tasks, $L$ is usually less than 50 in most cases while $D$ is 512 for base-scale models. Therefore, we can roughly get D >> L, and the computational cost of doubled source length is: $(2L)^2 * D + 2L * D^2$ =$2L * D * (2L + D)$ ≈ $2L * D * D$, about twice of baseline system: $L * D^2$.
>
> ②Empirically, we compare the training time of the baseline model and our model in the same environment. On 4 NVIDIA 2080Ti GPU, train both models on the WMT16 En-Ro dataset, baseline system updates 3.15 steps/s on average, and our model updates 1.58 steps/s on average.
>
> Although our model is computationally complex, it significantly improved translation quality by facilitating phrasal semantic information. Also, our "semantic only" setting model is less complex than the baseline system, but performs better. Moreover, we will try to diminish the gap between the "semantic only" model and the full model in future work.
>
> 8. **Question B**:
>
> We have claimed why we use these datasets in *Answer 5*. We do not expect our model to generalize well in low-resource languages since no special modification is made towards low-resource scenarios. Again, our primary motivation is to find and model phrases and subword-composed words. Therefore, our model may perform much better when there are more slangs in the corpus.
>
> 9. **Question C**:
>
> As mentioned in *Answer 3*, among all the non-contemporary related works, we could find no other strong SOTA better than UMST or perform similarly well.
>
> 10. **Question D**:
>
> Of course, every method will meet scenarios that it might not be optimal, and so will our model. As mentioned in the limitations section, if there's no semantic unit in the source sentence, our model will fall back into simply doubling the source sentence, and the performance may falter.
>
> Thanks again for your hard work in reviewing!

---

### Official Review · Reviewer_PSur · 2023-08-12

**Soundness:** 3

**Excitement:**

2: Mediocre: This paper makes marginal contributions (vs non-contemporaneous work), so I would rather not see it in the conference.

**Paper Topic And Main Contributions:**

The authors present an innovative method named "Attentive Semantic Fusion (AttnSF)" to address a recognized issue in Neural Machine Translation (NMT) – the potential overlooking of semantic connections between subwords. The paper suggests that by fusing the semantics of multiple subwords into a unified vector, the model can more adeptly comprehend the input sentence. A dual-layer architecture that utilizes both token-level and semantic-unit-level sentence information is proposed.

**Questions For The Authors:**

- Whether the proposed methods work in a larger-scale data or using a larger-scale model? do you have any ideas about the performance tredency when model or data scales up.
- Any relateness to the Multiword Expression, see https://direct.mit.edu/coli/article/43/4/837/1581/Multiword-Expression-Processing-A-Survey?
- Any impact on the decoding time?  Especially if semantic units could make decoded sequences shorter.
- Are there any other ways to implement the similar mechanism like semantic units? I am not an expert in translation, but I felt like there should be some similar mechanism like semantic units -- sorry, this is a little bit subjective since it seems using   semantic units is straightforward.

**Reasons To Accept:**

- Clarity: The concept of integrating multiple subwords into a single vector to encapsulate the meaning is explained lucidly.

- Experimental Results: It's commendable that the authors have provided experimental proof of the model's superiority over existing baselines. This gives weight to their proposal and ensures that it's not merely theoretical.



**Reasons To Reject:**

- Comparative Analysis: While the paper mentions outperformance over strong baselines, a more detailed comparative analysis would be beneficial. For instance, where does the baseline fail where AttnSF succeeds in translation?

**Reproducibility:**

3: Could reproduce the results with some difficulty. The settings of parameters are underspecified or subjectively determined; the training/evaluation data are not widely available.

**Reviewer Confidence:**

1: Not my area, or paper was hard for me to understand. My evaluation is just an educated guess.

---

> ### Author Rebuttal · Authors · 2023-08-29
>
> First of all, we have to express great gratitude to you for carefully reviewing the paper and providing the term **Multiword Expression(MWE)**. We believe its notion -*MWEs consist of several words (in the conventionally understood sense) but behave as single words to some extent*- largely overlaps with the notion of *semantic unit* in our work, and will add MWE in our Background section and illustrate the relationship between MWE and semantic unit.
>
> Now, we will answer your questions as follows:
>
> 1. **Comparative Analysis: Where does baseline fail where ASF succeeds?**:
>
> This kind of situation happens when some of the component words in MWE have their own meaning.
>
> Take a part of a sentence in WMT17 EN-ZH testset for example:
>
> *the striking lik@@ eness between British long jum@@ per Gre@@ g Ru@@ ther@@ ford and popular actor Neil Patrick Har@@ ris .*
>
> The output of baseline system is:
>
> 英国 长衣 选手 格雷格·鲁瑟福德 和 著名 演员 尼尔·帕特里克 · 哈里斯 之间 惊人 的 相似 。
>
> The output of out model is:
>
> 英国 跳远 运动员 格雷格·鲁瑟福德 ( Greg Rutherford ) 和 著名 演员 尼尔·帕特里克·哈里斯 ( Neil Patrick Harris ) 之间 的 惊人 相似性 。
>
> Here, *long jumper* means a kind of athlete, whereas *jumper* also means a kind of clothes. Our model treats *long jum@@ per* as a whole and correctly translates it into "跳远", while the baseline system fails to understand it and translates it into "长衣", meaning long clothing.
>
> 2. **Performance when model or data scales up**:
>
> We are sorry that with limited computing resources, we are not able to conduct sufficient experiments on larger model scales currently, nevertheless, we will leave this for future work. Intuitively, a conventional Transformer will still meet the discussed problem as model or data scales up, since it doesn't explicitly model semantic units. Therefore, the proposed ASF approach will still be effective, so far demonstrated by Base and big settings.
>
> Although we were not able to scale up the model size, we experimented on a larger dataset: the WMT14 En-Fr dataset, with 36M sentences. The results are as follows:
>
> | **dataset** | **method** | **model scale** | **multibleu** | **sacrebleu** | **chrf** | **comet** | **bleurt** |
> |:---:|:---:|:---:|:---:|:---:|:---:|:---:|:---:|
> | En-Fr | Transformer | base | 40.60| 37.38 | 63.62 | 84.03 | 70.00  |
> |  | Ours | base | **40.96** | **37.61** | **63.79** | **84.47** | **70.72**  |
>
> 3. **Any relatedness to Multiword Expression?**:
>
> Thanks for providing us with this survey, which we find quite helpful. The answer to your question is definitely yes, and the major notions of them are the same: several words/tokens behave like a single word to some extent.
>
> The difference mainly lies in the “domain of definition”: MWE describes the phenomenon in natural languages and every atom part is a word, whereas the semantic unit is defined on the model level and the atom part can be a subword. Another difference is semantic units have to consist of contiguous words, yet MWEs don’t have to. In future work, we may extend semantic units to token-level MWEs and benefit the research about MWEs.
>
> We will carefully revise our paper to introduce MWE in the background section and illustrate its relatedness to our work.
>
> 4. **Impact on decoding time? Can decoded sequences be made shorter?**:
>
> We are surprised that we have a similar idea, but for future work. In this model, the decoding time is slightly increased because the source sentence is prolonged. Empirically, we test the decoding speed of our model and the baseline system, by testing on the WMT14 En-De testset 5 times and calculating the average speed. This experiment is conducted on 1 Tesla T4 GPU. Our full model generates about 3377 tokens/s, and the baseline system generates about 3390 tokens/s.
>
> Your idea about shortening the decoded sequence is intriguing and may provide help for LLMs. Here is how we want to extend our work for this objective. Since we have verified the effectiveness of our ASF module, integrating the meaning of several tokens, we can further train a reverse module to decode the integrated representation into several tokens. In this case, Decoder takes in a long text and transfers it into a shorter “hidden text”, which can reduce the computing time, and dilates the “hidden text” into longer output. Therefore, we consider our work a first step toward this goal, which is quite meaningful.
>
> 5. **Other ways to implement similar mechanism like semantic units?**:
>
> Sorry, but we are a bit confused by the word “implement”. Please forgive our potential misunderstanding. Do you mean how to find the subwords that constitute a semantic unit? There are some parsing-based tools like *spaCy* or *Stanford Parser*, but they are either limited in extracting noun chunks or too slow. Or do you mean other mechanisms like MWE? We consider MWE a good mechanism to achieve our motivation, but the problem still lies in how to extract MWEs from texts. Just as you said, semantic units and WPE are straightforward. We will continue to study to find an elegant solution.
>
> Thanks again for your patience in reviewing and for providing the MWE mechanism. It lightens the way to possibly combine our work with existing linguistic phenomena in the future.

---

### Meta-Review · Area_Chair_VkaU · 2023-09-19

**Recommendation:** 4

**Metareview:**

This paper presents a novel approach to use semantics phrase-level information for Neural Machine Translation. The phrases are extracted with Word Pair Encoding (similarly to Byte Pair Encoding) and are integrated using an Attentive Semantic Fusion (ASF) by extending the network input.
The paper is clear and well written. The experiments are sound and the results show the beneficial impact of the method (although sometimes, rather low quantitative improvements are obtained).
In the rebuttal, the authors address most of the reviewers' comments. Their answers are convincing and, if all modifications are made, should lead to an improved version of the paper for the camera ready. This is much appreaciated and strengthen the submission.

---

### Decision · Program_Chairs · 2023-10-07

**Decision:**

Accept-Findings

**Comment:**

This paper presents a novel approach to use semantics phrase-level information for Neural Machine Translation. The phrases are extracted with Word Pair Encoding (similarly to Byte Pair Encoding) and are integrated using an Attentive Semantic Fusion (ASF) by extending the network input.
The paper is clear and well written. The experiments are sound and the results show the beneficial impact of the method (although sometimes, rather low quantitative improvements are obtained).
In the rebuttal, the authors address most of the reviewers' comments. Their answers are convincing and, if all modifications are made, should lead to an improved version of the paper for the camera ready. This is much appreaciated and strengthen the submission.